# The Effectiveness of Multi-Session FMT Treatment in Active Ulcerative Colitis Patients: A Pilot Study

**DOI:** 10.3390/biomedicines8080268

**Published:** 2020-08-03

**Authors:** Dorota Mańkowska-Wierzbicka, Marta Stelmach-Mardas, Marcin Gabryel, Hanna Tomczak, Marzena Skrzypczak-Zielińska, Oliwia Zakerska-Banaszak, Anna Sowińska, Dagmara Mahadea, Alina Baturo, Łukasz Wolko, Ryszard Słomski, Agnieszka Dobrowolska

**Affiliations:** 1Department of Gastroenterology, Dietetics and Internal Medicine, Poznan University of Medical Sciences, Heliodor Święcicki Hospital, Przybyszewskiego 49, 60-355 Poznań, Poland; marcingabryel1@o2.pl (M.G.); dagmaramahadea@gmail.com (D.M.); alina.baturo@gmail.com (A.B.); adobzach@ump.edu.pl (A.D.); 2Department of Biophysics, Poznan University of Medical Sciences, Grunwaldzka 6, 60-780 Poznań, Poland; stelmach@ump.edu.pl; 3Department of Dermatology and Venereology, Poznan University of Medical Sciences, Przybyszewskiego 49, 60-355 Poznań, Poland; tomczak.hanna@spsk2.pl; 4Central Microbiological Laboratory, Święcicki University Hospital, Przybyszewskiego 49, 60-355 Poznań, Poland; 5Institute of Human Genetics, Polish Academy of Sciences, Strzeszyńska 32, 60-479 Poznań, Poland; mskrzypczakzielinska@gmail.com (M.S.-Z.); o.zakerska.banaszak@gmail.com (O.Z.-B.); ryszard.slomski@up.poznan.pl (R.S.); 6Department of Computer Science and Statistics, Poznan University of Medical Sciences, Rokietnicka 7, 60-806 Poznań, Poland; ania@ump.edu.pl; 7Department of Biochemistry and Biotechnology, University of Life Sciences, Dojazd 11, 60-632 Poznań, Poland; lukwolko@gmail.com

**Keywords:** fecal microbiota transplantation (FMT), inflammatory bowel disease (IBD), ulcerative colitis (UC), gut microbiota

## Abstract

The modification of the microbiome through fecal microbiota transplantation (FMT) is becoming a very promising therapeutic option for inflammatory bowel disease (IBD) patients. Our pilot study aimed to assess the effectiveness of multi-session FMT treatment in active ulcerative colitis (UC) patients. Ten patients with UC were treated with multi-session FMT (200 mL) from healthy donors, via colonoscopy/gastroscopy. Patients were evaluated as follows: at baseline, at week 7, and after 6 months, routine blood tests (including C reactive protein (CRP) and calprotectin) were performed. 16S rRNA gene (V3V4) sequencing was used for metagenomic analysis. The severity of UC was classified based on the Truelove–Witts index. The assessment of microbial diversity showed significant differences between recipients and healthy donors. FMT contributed to long-term, significant clinical and biochemical improvement. Metagenomic analysis revealed an increase in the amount of *Lactobacillaceaea*, *Micrococcaceae*, *Prevotellaceae*, and *TM7 phylum*sp.*oral clone EW055* during FMT, whereas *Staphylococcaceae* and *Bacillaceae* declined significantly. A positive increase in the proportion of the genera *Bifidobacterium*, *Lactobacillus*, *Rothia*, *Streptococcus*, and *Veillonella* and a decrease in *Bacillus*, *Bacteroides*, and *Staphylococcus* were observed based on the correlation between calprotectin and *Bacillus* and *Staphylococcus;* ferritin and *Lactobacillus*, *Veillonella,* and *Bifidobacterium* abundance was indicated. A positive change in the abundance of *Firmicutes* was observed during FMT and after 6 months. The application of multi-session FMT led to the restoration of recipients’ microbiota and resulted in the remission of patients with active UC.

## 1. Introduction

Inflammatory bowel disease (IBD) is a chronic progressive and idiopathic inflammatory disorder of the gastrointestinal tract (GI) tract, which includes ulcerative colitis (UC) and Crohn’s disease (CD). Although the etiology of IBD remains largely unknown, it involves a complex interaction between the genetic, environmental, and aberrant immune responses [1]. In recent times, the involvement of gut microbiota is considered a new probable factor strongly connected with IBD pathogenesis [2]. However, it is still unclear whether this dysbiosis is a primary or secondary event in the relationship with IBD. In this regard, Tomasello et al. [3] reported several mechanisms that are responsible for gut dysbiosis. Among the various etiopathogenic hypotheses, the most important one suggests that change in the saprophytic microbial flora causes mucosal damage. Modifications of intracellular tight junctions cause significant penetration of antigens, which leads to activation of the intestinal lymphatic system (MALT) [4]. Moreover, commensal microbiota plays a crucial role in the regulation of intestinal immune homeostasis. For example, *Bacteroides fragilis* through the action of the bacterial-derived polysaccharide A (PSA) affects systemic Th1 response [5]. Additionally, intestinal microbiota composition is regulated by specialized ileal epithelial cells. Paneth cells act by secreting granule contents that include antimicrobial peptides (secretory phospholipase A2 and α-defensins), thus any defects in the autophagy pathway lead to cell pathology [6]. So far, previous studies have presented inconsistent and inconclusive results on alterations in the intestinal microbiota in IBD patients. The gut microbial composition of patients with IBD is characterized by lower taxonomic levels of the most frequently assessed beneficial bacteria, such as *Faecalibacterium prausnitzii* and *Roseburia intestinalis* [7]. These microbes are responsible for butyric acid production, and as a result, can protect and modulate the intestinal immune response. Additionally, UC patients’ microbiota may harbor adherent/invasive *Escherichia coli* and *Shigella* species of the *Enterobacteriaceae* family [8]. Nevertheless, some changes in microbiota composition are observed in CD patients only, with no further confirmation in UC patients (i.e., in *Actinobacteria, Firmicutes*, *Proteobacteria*, *Bacteroidetes*, and *Verrucomicrobia)* [9,10,11,12,13], which makes the clinical investigation of microbiota in this group of patients even more complicated. Currently, one of the recent therapeutic options for UC patients, fecal microbiota transplantation (FMT), is applied in the clinical setting with a focus on restoring dysbiosis. FMT is an infusion of stool from a healthy donor into the colon, or its delivery through the upper gastrointestinal tract, to a recipient with a disease believed to be related to an unhealthy gut microbiome [14]. Recent studies [15,16,17,18,19] have proven the efficacy of FMT in the treatment of *Clostridium difficile* infection, with cure rates up to 90%. From a long-term perspective, it might be possible that FMT will be useful in the maintenance of IBD remission, which is of high importance in this treatment. 

## 2. Materials and Methods

### 2.1. Study Design and Patients

This was a clinically controlled intervention study conducted at the Department of Gastroenterology of Heliodor Swiecicki Clinical Hospital, at Poznan University of Medical Sciences between January 2018 and December 2019. The study protocol was approved on 10 November 2016 by the Research Ethical Committee of Poznan University of Medical Sciences in Poland (No. 1004/16) and followed the requirements of the Declaration of Helsinki. All patients (donors and recipients of FMT) provided written consent.

Ten patients participated in the study, where multi-session FMT treatment was performed in weeks 1–6. Simultaneously, at the baseline, after the 6th course of FMT administration (week 7), and at the end of the follow-up period (6 months), stool sample analyses were performed to assess the changes in the microbiome and the concentration of fecal calprotectin. Six healthy stool donors were recruited from volunteers who reported to Heliodor Swiecicki Clinical Hospital in Poznan. Finally, 42 fecal samples collected from donors and recipients were used for microbial profiling.

### 2.2. Characteristics of FMT Recipients

Eligible patients were adults (aged over 18) with moderately to severely active UC (Truelove–Witts Severity Index 2–3), with no previous FMT history. The prevailing body mass index (BMI) of enrolled patients was 22 ± 2.75 kg/m^2^. UC was diagnosed based on clinical, endoscopic, and histological criteria. Exclusion criteria were as follows: colonic surgery, gastrointestinal infection including parasitic and *C. difficile* infections, indeterminate colitis, irritable bowel syndrome, food allergy, co-morbid chronic disease, history of cancer, pregnancy, use of antibiotics or probiotics during the 3 months before enrolment. During the study, concomitant treatments using 5-aminosalicylic acid immunomodulators were permitted, as long as the dose was stable for 4 weeks, and oral corticosteroids with the mandatory taper of 4–5 mg per week. However, neither the use of antibiotics nor probiotics was allowed.

### 2.3. Characteristics of Fecal Donors

Fecal donors were healthy individuals with regular Body Mass Index (BMI: 18.5–24.9 kg/m^2^), who consumed a diet following the principles of proper nutrition, were unrelated to recipients, and underwent detailed tests to rule out some diseases. A detailed medical history interview, physical examination, body composition analysis, and evaluation according to the following criteria were performed [20,21]. Following the guidelines by Cammarota et al. [22], the exclusion criteria for fecal donors were as follows: the presence of HIV, hepatitis B and C virus, cytomegalovirus, Epstein–Barr virus, toxoplasmosis, syphilis, *Clostridium difficile, Yersinia* spp., *Campylobacter jejuni Shigella* spp., *Helicobacter pylori*, parasites and potential carriers of methicillin-resistant *Staphylococcus aureus* (MRSA), vancomycin-resistant *Enterococcus* (VRE), extended-spectrum-β-lactamase (ESBL) producing bacteria, carbapenem-resistant *Enterobacteriaceae* (CRE), New Delhi metallo-β-lactamase (NDM), *Klebsiella pneumoniae carbapenemase* (KPC), and OXA-48. An additional exclusion criterion was the use of antibiotic therapy in the last 6 months. Basic biochemical tests (CRP, TSH, creatinine) and complete blood count (CBC) were performed. All samples were stored at −80 °C immediately after collection. Donors’ fecal samples were examined twice in the pre-screening period and 6 weeks later to minimize the risk of disease transmission.

### 2.4. Fecal Sample Preparation and FMT Intervention

From the collection of feces to the preparation of the transplant, strict procedures were followed to ensure sterile conditions. In the filtration process, the material was deprived of food residues; 180 mL of the mixture was taken successively, poured into disposable sterile sealed bottles, and then 20 mL of sterile glycerol as a cryoprotectant was added to achieve the mixture of feces: saline: glycerol in a ratio of 2:7:1. The bottles were successively frozen at −80 °C to ensure that the value of the material was identical to fresh material. On the day of transplantation, the material was thawed according to the procedure. The bottles with the material were additionally wrapped with aluminum foil to minimize the harmful effects of light and provide the patient with psychological comfort during the procedure. Intestinal microbiota transplantation was performed via colonoscopy or gastroduodenoscopy issues. Participants received loperamide 2 mg orally to reduce intestinal peristalsis and to maintain the suspension for a longer period. Fifty grams of fecal material was diluted with 150–200 mL sterile saline solution (0.9%). Infusion of the fecal suspension was ongoing via the biopsy channel of the flexible endoscope using the biopsy channel cap with extension tubing over the course of 5 min followed by 100 mL of physiological saline. The first dosage of FMT was given during the colonoscopy approach, throughout which 200 mL of a fecal suspension of donor stool was injected into the cecum. Additionally, during colonoscopy, the severity and extent of UC were estimated. Five subsequent sessions of FMT were performed via the upper GI tract route once a week for five consecutive weeks. A volume of 200 mL of fecal suspension was infused into the descending part of the duodenum. No side effects were reported.

### 2.5. Metagenomic Analysis and Data Processing

#### 2.5.1. Sample Collection and DNA Extraction

Stool samples from donors and recipients (before and 1 month and 6 months after FMT) were immediately stored at −80 °C until genetic material isolation. From an approximately 200 μL feces aliquot, the bacterial DNA was extracted by using a Genomic Mini AX Bacteria+ Kit (A&A Biotechnology, Gdansk, Poland) according to the manufacturer’s protocol.

#### 2.5.2. 16. S rRNA Gene Amplification and Sequencing

16S rRNA gene variable regions V3 and V4 were amplified in a two-step barcoding approach according to the 16S Metagenomic Sequencing Library Preparation protocol (https://support.illumina.com/downloads/16s_metagenomic_sequencing_library_preparation.html, Part# 15044223Rev.B, Nov 27, 2013) using Illumina primers with overhang adapters (Forward: 5′TCGTCGGCAGCGTCAGATGTGTATAAGAGACAGCCTACGGGNGGCWGCAG3′, Reverse: 5′GTCTCGTGGGCTCGGAGATGTGTATAAGAGACAGGACTACHVGGGTATCTAATC3′), indices (Nextera^®^ XT Index Kit 96, Illumina, San Diego, CA, USA), and KAPA HiFi HotStart ReadyMix PCR Kit (KAPA Biosystems, Cape Town, South Africa). Library concentration and quality were assessed on the Qubit^®^ 2.0 Fluorometer (Thermo Fisher Scientific, Inc., Waltham, MA, USA) and the Agilent 2100 Bioanalyzer System (Agilent, Santa Clara, CA, USA). Subsequently, an 8 pM pooled library was enriched with a 20% PhiX Control v3 (Illumina, San Diego, CA, USA), and paired-end sequencing was performed on the Illumina MiSeq platform using the MiSeq Reagent Kit v3 (600 cycles).

### 2.6. Biochemical Assessment

In order to monitor the patient’s clinical features, routine diagnostic blood tests were performed including a hemoglobin count (Hb (g/L)), white cell count (WBC (10^9^/L)), red blood count (RBC (10^12^/L)), platelet count (PLT (10^9^/L)), markers of inflammation such as C reactive protein (CRP (mg/dL)), iron (Fe (ug/dL)), total iron-binding capacity (TIBC (ug/dL)), serum ferritin (ug/L) as an indicator of iron storage, total protein level (TP g/dL), and albumin (g/dL). Additionally, fecal calprotectin as a non-invasive test for the direct evaluation of intestinal inflammation was measured using a commercially available quantitative enzyme-linked immunoassay (PhiCal Calprotectin Elisa Kit, Immunodiagnostic, Bensheim, Germany). Blood samples were taken after 14 h of fasting. Biochemical assessments and metagenomic analysis were performed in a certified laboratory according to standardized procedures and good laboratory practice at the baseline (week 0), after the 6th course of therapeutic intervention, and at the end of the follow-up study.

### 2.7. Statistical Analysis

Obtained sequencing reads were processed for quality check, demultiplexing, trimming, and alignment using CLC Genomic Workbench 8.5 and CLC Microbial Genomics Module 1.2. (Qiagen Bioinformatics, Aarhus, Denmark). Chimeric sequences were removed and operational taxonomic units (OTUs) were clustered against the SILVA v119 97% 16S rDNA gene database, following Quast et al. [23].

The results are presented as percentage average with 1 standard deviation for normally distributed continuous variables, or median (interquartile range) for non-normally distributed continuous variables as tested by the Shapiro–Wilk test. A *p*-value of less than 0.05 was considered significant. The associations between clinical parameters and bacteria were tested by the Spearman rank coefficient. For comparison of OTU richness in all studied groups, Kruskal–Wallis (*p* = 0.0013) and U Mann–Whitney tests were used. Comparisons between two groups were analyzed by the U Mann–Whitney test, and for more than two groups by ANOVA with post hoc Tukey test or Friedman with post hoc Dunn test for normally distributed continuous variables or non-normally distributed variables, respectively. Statistical analysis was performed using Dell Inc. (2016). Dell Statistica (data analysis software system), version 13. software.dell.com.

## 3. Results

The clinical characteristics of patients are presented in Table 1. The diversity and richness of the fecal microbiota were significantly higher in donors compared with UC patients. At week 7 after FMT, the diversity and richness of the recipients’ fecal microbiota increased but then slightly decreased after 6 months (Figure 1). Assessment of phylum microbial diversity between recipients and donors indicated statistically significant changes in *Verrucomicrobiota* (Figure 2). Metagenomic analysis also indicated statistical changes in the family diversity of *Christensenellaceae, Family XIII, Ruminococcaceae,* and *Verrucomicrobiaceae* (Figure 3A). Within the range of genera, significant changes between donors’ and recipients’ microbiota were observed in the numbers of *Akkermansia*, *Alloprevotella, Christensenella, Coprococcus, Dorea, Faecalibacterium, Incertae Sedis-03, Incertae Sedis-04, Incertae Sedis-06, Ruminococcus*, and *Subdoligranulum* (Figure 4A).

Positive changes in the abundance of the phylum *Firmicutes* over the course of FMT were indicated in recipient patients, where a statistically significant increase was observed between baseline and the end of the follow-up period (6 months) (Figure 5). Metagenomic analysis indicated a significant increase in the family abundance of *Lactobacillaceaea, Micrococcaceae*, *Prevotellaceae*, and *TM7 phylum* sp. *oral clone EW055* during FMT, whereas *Bacillaceae* and *Staphylococcaceae* declined significantly (*p* < 0.05) (Figure 3B). Moreover, positive, significant changes in the genus diversity were shown in the abundance of *Anaerococcus*, *Bifidobacterium*, *Lactobacillus*, *Rothia, TM7 phylum* sp. *oral clone EW055*, and *Veillonella.* In contrast, we observed a decreased richness of *Bacillus, Bacteroides*, and *Staphylococcus* after FMT and 6 months later, while *Streptococcus* decreased after FMT but increased at 6 months post FMT (Figure 4B).

Biochemical results are shown in Table 2. The disease activity was determined based on clinical and biochemical indicators. The comparison of biomarkers such as red blood count (RBC), hematocrit HCT, total iron-binding capacity (TIBC), C reactive protein (CRP), total protein level (TP), and calprotectin over the three time points of analysis (baseline, 7 weeks, and 6 months) showed statistically significant and clinically beneficial changes (*p* < 0.05). These significant, positive changes in biomarker concentrations were maintained during the follow-up period (6 months after FMT) for all of them. However, directly after FMT (at the 7th week), statistically significant changes were observed only for CRP and TP. Disease activity was classified based on the Truelove and Witts score (*p* < 0.05), which was also significant in the follow-up period. Additionally, before FMT administration we observed a positive correlation in recipients’ stool between calprotectin level and *Bacillus* and *Staphylococcus* genera, and *Staphylococcaceae* family richness. Furthermore, we noted that low serum ferritin concentration coexisted with decreased abundance of the *Lactobacillaceae* family, *Lactobacillus* genera, and *Veillonella* species, whereas augmented *Bifidobacterium* in recipients’ stool 6 months after FMT contributed to an increase in the level of serum ferritin (Table 3).

## 4. Discussion

In the current study, the effectiveness of multi-session FMT was proven as a promising therapeutic option for moderately to severely active UC patients. The high percentage of patients (60%) that achieved clinically significant improvement highlights the importance of FMT in long-term microbial diversity modulation in UC patients. No serious adverse effects were observed. Therefore, FMT can be characterized as a safe procedure in the clinical setting.

There are several factors such as delivery route, frequency, dosage, and microbiota resources that may significantly affect the outcome of FMT in UC patients. The most adequate route of administration for FMT remains to be defined. Multiple approaches to FMT including upper-gut delivery (oral capsules of fecal microbiota), mid-gut delivery (naso-intestinal tubes, gastroduodenoscopy, transendoscopic enteral tubing (TET), intestinal stoma, percutaneous endoscopic gastrostomy with jejunal tubes (PEG-J)), or lower-gut delivery (colonoscopy, enema, infusions through colonic stomas and a new delivering technic colonic transendoscopic enteral tubing (TET) or combinations) have been described in several studies [24,25,26,27]. Moayyedi et al. [28] performed FMT via enema, and Rossen et al. [29] used a nasoduodenal tube. The numerous adverse effects noted with the use of the naso-intestinal route convinced us that FMT should be performed using colonoscopy and gastroscopy [30]. The same approach was followed by Goyal et al. [31]. It has to be mentioned that the intensity and duration of FMT may also affect the outcome. In the current study, patients received fecal material once weekly for six consecutive weeks. In another study [32], stool was administered five days per week for eight weeks. Rossen et al. [29] delivered fecal material at weeks 0 and 3. Clinical improvement was observed in cases with higher frequency and longer duration of FMT, thus indicating this approach as superior to rare and short-term interventions, in which no significant differences in clinical and endoscopic remission in UC patients were observed [29]. In general, greater intensity and duration of FMT result in a higher rate of remission in patients with UC. Moreover, the efficacy of FMT in UC patients is influenced by the taxonomic composition of the donor’s intestinal microbiota.

According to the literature published so far, no statistically significant differences were found when fresh stools were used for FMT as compared to frozen-thawed fecal material [33]. However, it has not been established whether single donor vs. multi-donor stool has better efficacy in re-establishing enteric homeostasis. It would seem that multi-donor stools may present with greater microbial diversity. It should be highlighted that microbial diversity also plays a crucial role in preventing the overgrowth of pathogens in UC. Therefore, donors chosen for the study were healthy individuals with a beneficial microbiota composition, without any co-morbidities. The metagenomics analysis indicated statistically significant differences between donors and recipients in the family diversity of *Christensenellaceae, Family XIII, Ruminococcaceae,* and *Verrucomicrobiaceae*, which could directly influence changes in specific genus microbial diversity. As shown in the meta-analysis by Mancabelli et al. [34], *Christensenellaceae* is an indicator of a healthy gut. Indeed, a growing body of evidence supports the consistent reduction of *Christensenellaceae* in patients with UC [9,35] and Crohn’s disease [9]. *Christensenellaceae* is a recently described family associated with a lean BMI and short-chain fatty acids (SCFAs) [36,37,38]. A noteworthy phylum is the *Verrucomicrobia*. It is a widely held view that reduced levels of *Verrucomicrobia* in the gastrointestinal tract are indicative of gut dysbiosis and are observed in UC patients [39,40,41]. This is consistent with our metagenomic analysis, in which microbiota of healthy donors revealed a relatively higher abundance of *Verrucomicrobia*. Alam et al. [42] reported in a mouse model that *Akkermansia muciniphila* (which belongs to the *Verrucomicrobia* phylum) plays a major role in intestinal wound healing and is considered to be a “probiont” species that contributes towards the repair of mucosal wounds. Furthermore, *Akkermansia* is associated with the immune response and the induction of interleukin-10 (IL-10) cytokine and SFCA production, supplying energy to goblet cells that produce mucin [43]. Patients with refractory UC, who achieved remission after FMT had a significantly higher relative abundance of *Ruminococcaceae* including *Ruminococcus* spp. and *Akkermansia muciniphila* [38]. This could be due to their ability to process nondigestible carbohydrates and promote the production of SFCAs as well. According to Paramsothy et al. [44], increased abundance of *Bacteroides* (*B. fragilis* and *B. finegoldii*) in donor stool may be associated with observed remission in patients receiving FMT, while changes in *Streptococcus* may not be related to FMT response. Santoru et al. [10] also showed that the microbial profile is characterized by a significantly higher presence of *Verrucomicrobia, Firmicutes, Proteobacteria,* and *Fusobacteria* in the IBD group. Additionally, bacterial taxa such as *Coprococcus*, *Mucispirillum*, *Odoribacter*, *Prevotella*, and *Oscillospira* were increased in abundance specifically in the early regenerative mucosa [42].

Six months after FMT, a rise in the abundance of the *Micrococcaceae* family in the recipients’ stools was observed. Walujkar et al. [45] showed an increase in the abundance of the genus *Micrococcus* in mucosal-associated microbiota (MAM) of exacerbated UC patients when compared with the remission phase. This was in the line with a study by Kiernan et al. [46] where a lower abundance of *Actinobacteria* phyla including *Mycobacteriaceae*, *Micrococcaceae*, and *Streptomycineae* was detected in healthy individuals. In the current study, an increase in the *Micrococcaceae* family may be an indication that microbiota composition is going to change in the pro-inflammatory direction, and from a long-term perspective, FMT will be needed in UC patients as a regular repeated procedure. Chen et al. [47] reported changes in the abundance of *Faecalibacterium prausnitzii* following FMT, which is of high importance for inflammatory diseases. In the intestine, *F. prausnitzii* produces butyrate, which can reduce intestinal mucosa inflammation by inhibiting the nuclear factor kappa-light-chain-enhancer of activated B cells (NF-κB) transcription factor [48]. Another study [49] showed differences in the abundance of bacteria such as *Lachnospiraceae, Bacteroidetes*, *Proteobacteria*, and *Clostridium clusters IV* and *XIVa (Firmicutes)* in UC patients after FMT. This is in line with the current study, where beneficial changes in the abundance of the phylum *Firmicutes* over the course of FMT were indicated in recipient UC patients. Dutta et al. [50] also reported, during FMT, a significant rise in *Lachnospiraceae*, which have anti-inflammatory qualities. Data concerning the *Bacteroidetes* phylum are not unambiguous. According to our baseline data, a relatively higher abundance of *Bacteroides* was noted in recipients compared to donors, and it declined after FMT. Some studies [51,52,53] reported a decrease in the abundance of the *Bacteroides* in Crohn’s disease and UC patients. In contrast, Andoh et al. [54] reported an increased abundance of this phylum in IBD patients.

Analysis of our recipients’ stools indicated a significantly different abundance of *Bifidobacterium* and *Lactobacillus* genera, comparing the pre and post-fecal microbiota transplantation samples and at 6 months after FMT. The *Lactobacilli* and *Bifidobacteria* are responsible for the process of dietary carbohydrate fermentation in the colon, leading to the production of SCFAs such as butyrate, propionate, and acetate [55,56,57,58]. Due to their potential therapeutic anti-inflammatory components, some strains of Lactobacilli might positively influence the activity of inflammation in IBD patients. According to Nishida et al. [59], an increasing proportion of *Bifidobacterium* and Lactobacillus in recipients’ stools might contribute to a favorable response to FMT. Nevertheless, some studies [60,61,62] conducted in IBD patients have shown a significant depletion of beneficial Lactobacillus in both UC and CD individuals. At the baseline of our study, an overabundance of *Staphylococcus* taxa was observed in recipients, and it was consistently depleted after FMT. The mechanism of colonization of *Staphylococcus* in the human gut is still barely understood. However, some studies suggest an association of *Staphylococcus aureus* with IBD [63,64].

The *Staphylococcus* genus was found more frequently in UC patients with coexisting arthritis [65]. Furthermore, we observed an increased abundance of the *Prevotellecaeae* family in our recipients’ microbiota concomitantly after FMT and 6 months later. A similar observation was presented by Paramsmothy et al. [32], showing that *Prevotella* was a prominent feature in several patients during and after fecal microbiota transplantation. Several studies [66,67,68] have shown associations between the Mediterranean diet and an increased abundance of *Prevotella*.

From the clinical point of view, there are also several biomarkers such as C reactive protein (CRP), erythrocyte sedimentation rate (ESR), platelets, and fecal calprotectin that can be used for objective evaluation of disease activity and inflammation in IBD patients [69]. In the current study, the concentration of CRP and calprotectin significantly improved directly after FMT (*p* < 0.05), and the effect was maintained in the follow-up period of observation. Similarly to our results, Uygun et al. [70] reported that CRP and sedimentation level improved after FMT in both responder and non-responder to this therapy groups of patients. Comparably, Cold et al. [71] in an open-labeled pilot study confirmed that median fecal calprotectin decreased significantly during the FMT treatment period, but increased again in the follow-up. Additionally, we observed a positive correlation between calprotectin level and *Bacillus* and *Staphylococcus* richness in recipients’ stools before FMT administration. Calprotectin belongs to the antimicrobial proteins that inhibit the growth of microbial species [72,73]. Damo et al. [74] reported that calprotectin hinders the growth of bacterial pathogens such as *Staphylococcus aureus*, *S. epidermidis*, *S. lugdunesis*, and *Enterococcus fecalis,* etc. Furthermore, we noticed that low serum ferritin concentration coexisted with decreased abundance of *Lactobacillus* and *Veillonella* species, whereas augmented *Bifidobacterium* in recipients’ stools 6 months after FMT contributed to an increased level of serum ferritin. One third of IBD patients suffer from iron deficiency anemia (IDA) or anemia of chronic disease (ACD). Das et al. [75] showed that gut microbial metabolites may regulate host systemic iron homeostasis via ferritin regulation. Lactobacillus species are the crucial bacteria in ascertaining intestinal iron levels and attenuating host iron absorption.

This study has some limitations. The small number of patients who enrolled in this study means that it should be treated as a pilot in the further evaluation of FMT efficacy in UC patients, where a control group should be introduced to avoid potential bias in the data analysis.

## 5. Conclusions

FMT seems to be a promising therapy for the management of moderately to severely active UC. Our pilot study demonstrated that six rounds of intensive, weekly FMT administration contributed to clinical (Truelove and Witts score) and biochemical (CRP, calprotectin) improvement not only immediately after FMT, but also persisting up to 6 months of follow-up. Metagenomic analysis revealed significant differences in microbial diversity and richness between recipients and donors. After FMT, we observed a positive increase in the amount of *Lactobacillaceaea*, *Micrococcaceae*, *Prevotellaceae*, and *TM7 phylum* sp. *oral clone EW055*, whereas *Staphylococcaceae* and *Bacillaceae* declined significantly. Furthermore, we revealed a positive change in the abundance of Firmicutes both during FMT and after 6 months. It seems that the efficacy of this study might be related to the good donor microbial characteristics and the planned scheme and route (one colonoscopy and five rounds of gastroduodenoscopy) of FMT.

## Figures and Tables

**Figure 1 biomedicines-08-00268-f001:**
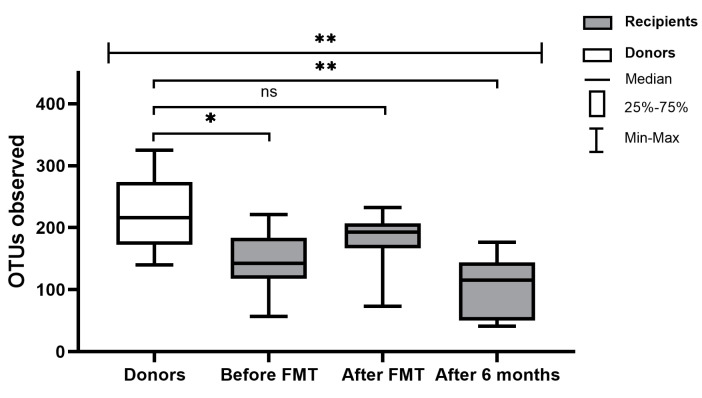
Gut microbial diversity in terms of operational taxonomic unit (OTU) richness in donors and ulcerative colitis (UC) patients in the peri-fecal microbiota transplantation (FMT) period. * *p* < 0.05, ** *p* < 0.01, ns: no significance.

**Figure 2 biomedicines-08-00268-f002:**
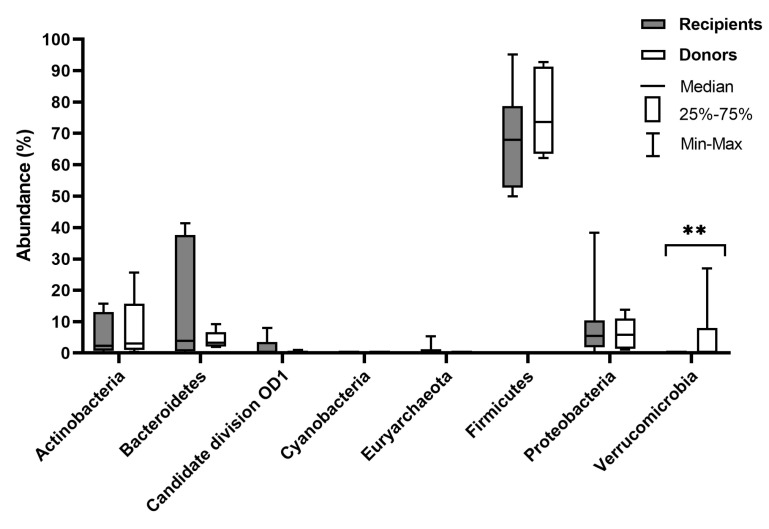
Comparison of microbial composition at the phylum level between recipients and donors. ** *p* < 0.005.

**Figure 3 biomedicines-08-00268-f003:**
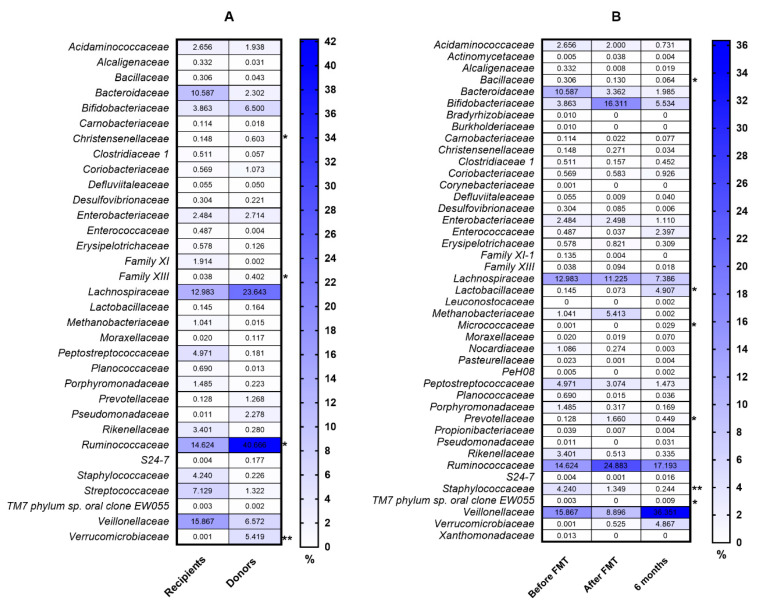
Comparison of microbial composition at the family level between recipients and donors (**A**) and in UC patients in the peri-FMT period (**B**). * *p* < 0.05, ** *p* < 0.01.

**Figure 4 biomedicines-08-00268-f004:**
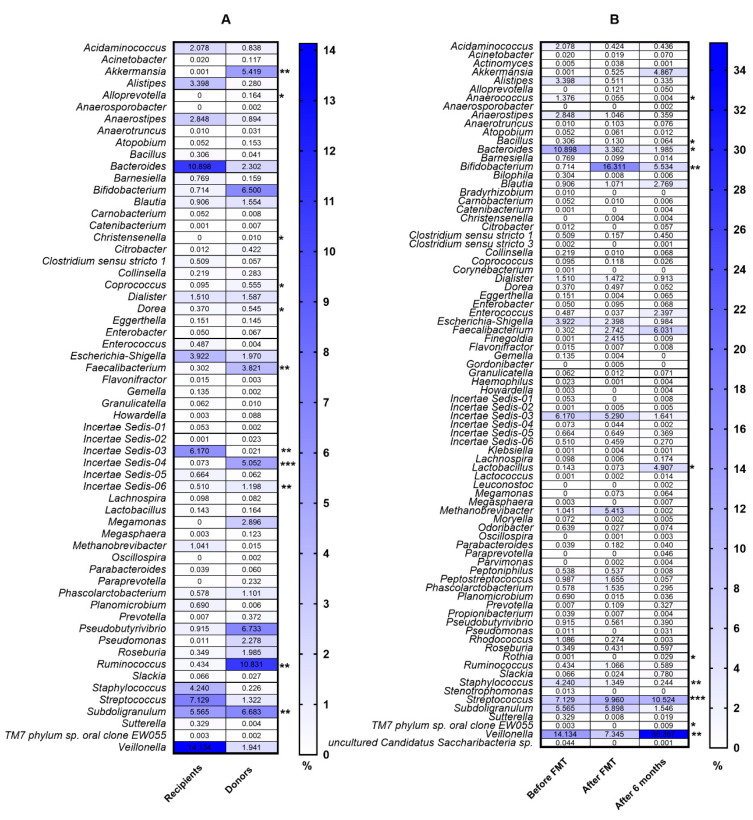
Comparison of microbial composition at the genus level between recipients and donors (**A**) and in UC patients in the peri-FMT period (**B**). The numbers in rectangles represent the abundance for each detected bacterial genus in %. Statistical significance: * *p* < 0.05, ** *p* < 0.01, *** *p* < 0.001.

**Figure 5 biomedicines-08-00268-f005:**
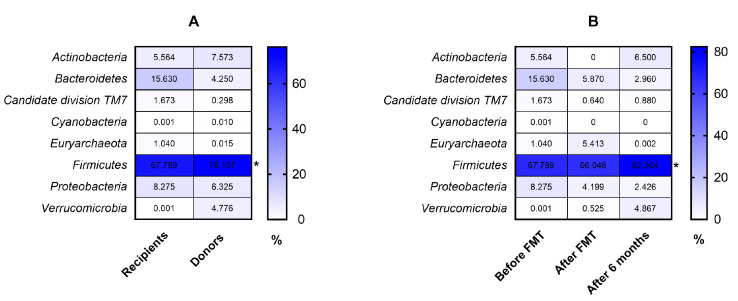
Comparison of microbial composition at the phylum level between recipients and donors (**A**) and in UC patients in the peri-FMT period (**B**). The numbers in rectangles represent the abundance for each detected bacterial phylum in %. Statistical significance: * *p* < 0.05.

**Table 1 biomedicines-08-00268-t001:** Baseline patient characteristics.

Analyzed Variable	N = 10
Age (years)	47.5 ± 18.16
Disease duration (years)	5.9 (3−10)
BMI (kg/m^2^)	22 ± 2.75
Male sex	5
Disease extent	
-Proctitis	0
-Left-sided colitis	4
-Extensive colitis	4
-Pancolitis	2
Medical treatment *	
-Oral 5-ASA	10
-Oral steroids	4
-Immunomodulators	4
Nonsmoker	6

***** Aminosalicylates: mesalazine, sulfasalazine; corticosteroids: prednisone, methylprednisolone; immunomodulators: azathioprine, 6-mercaptopurine.

**Table 2 biomedicines-08-00268-t002:** Biochemical results before and after FMT and after 6 months.

Analyzed Parameter	n	Before FMT	After FMT	After 6 Months	*p*-Value
WBC (10^3^/L)	10	22.40 (20–24)	7.15 (5.2–9.4)	7.2 (6.2–9.4)	0.904 ^b^
RBC (10⁶/L)	10	3.97 ± 0.62	4.11 ± 0.63	4.438 ± 0.46	0.029 ^a^
					Before vs. After 0.675
					Before vs. After 6mo 0.027
					After vs. After 6mo 0.141
HGB (g/dL)	10	11.61 ± 2.38	12.18 ± 1.88	12.63 ± 1.92	0.099 ^a^
HCT (%)	HCT	36 (2.7–39)	36.65 (33.6–39.5)	40.35 (35.5–42.7)	0.049 ^b^
					Before vs. After 0.281
					Before vs. After 6mo 0.057
					After vs. After 6mo 1.000
MCV (fL)	10	86.75 (84–88)	86.35 (84.9–88)	85.35 (84.9–87)	0.905 ^b^
RDW-CV (fL)	10	12.55 (12.3–13.7)	13.7 (12.1–16)	13.85 (12.8–14.5)	0.283 ^b^
PLT (10^3^/L)	10	313.4 ± 128.17	344.7 ± 118.83	299.1 ± 104.82	0.059 ^a^
Iron (µg/dL)	10	48.1 ± 24.6	51.7 ± 22.92	71.1 ± 39.75	0.075 ^a^
TIBC (µg/dL)	10	243.2 ± 72.81	278.6 ± 36.81	316.5 ± 45.01	0.004 ^a^
					Before vs. After 0.171
					Before vs. After 6mo 0.003
					After vs. After 6mo 0.135
CRP (mg/L)	10	9.5 (7.7–82)	5.2 (3.2–5.7)	3.4 (2.4–7.4)	0.0004 ^b^
					Before vs. After 0.011
					Before vs. After 6mo 0.0004
					After vs. After 6mo 1.000
Ferritin (ng/mL)	10	33 (23–50)	35.5 (19–42)	37 (24–44)	0.388 ^b^
TP (g/dL)	10	6.7 ± 0.45	7.18 ± 0.51	7.325 ± 0.52	0.001 ^a^
					Before vs. After 0.013
					Before vs. After 6mo 0.002
					After vs. After 6mo 0.604
ALB (g/dL)	10	3.72 ± 0.65	4.06 ± 0.28	4.196 ± 0.33	0.127 ^a^
Calprotectin (µg/g)	10	1500 (969–1590)	1095 (1000–1280)	510 (91–800)	0.002 ^b^
					Before vs. After 0.221
					Before vs. After 6mo 0.001
					After vs. After 6mo 0.221
Disease activity (Truelove and Witts Severity Index)	10	3 (3–3)	2 (2–2)	1 (1–2)	0.0001 ^b^
					Before vs. After 0.016
					Before vs. After 6mo 0.0003
					After vs. After 6mo 0.791

^a^ ANOVA post hoc Tukey, ^b^ Friedman post hoc Dunn.

**Table 3 biomedicines-08-00268-t003:** Selected biochemical parameters and bacteria, before and after FMT and after 6 months.

Phylum									
Selected Bacteria	Before FMT			After FMT			After 6 Months		
	CRP	Ferritin	CALPR *	CRP	Ferritin	CALPR *	CRP	Ferritin	CALPR *
*Firmicutes*	0.405 (0.297)	0.947 (0.024)	0.575 (−0.202)	0.382 (0.311)	0.556 (−0.212)	0.275 (−0.383)	0.511 (−0.236)	0.345 (−0.334)	0.128 (−0.515)
**Family**									
*Lactobacillaceae*	0.851 (0.068)	0.035 (−0.668)	0.427 (−0.283)	0.228 (−0.419)	0.570 (0.205)	0.903 (0.045)	0.379 (0.313)	0.852 (−0.068)	0.672 (0.153)
*Staphylococcaceae*	0.336 (0.340)	0.662 (0.159)	0.001 (0.861)	0.353 (0.329)	0.059 (0.612)	0.097 (0.553)	0.211 (0.433)	0.293 (0.370)	0.131 (0.511)
**Genus**									
*Anaerococcus*	0.793 (−0.096)	0.756 (0.113)	0.97 (−0.014)	0.227 (−0.42)	0.141 (−0.5)	0.918 (−0.037)	0.416 (0.29)	0.415 (0.291)	0.416 (0.29)
*Bacillus*	0.135 (0.506)	0.424 (0.285)	0.003 (0.829)	0.021 (0.713)	0.947 (−0.024)	0.973 (0.012)	0.173 (0.467)	0.102 (0.547)	0.65 (0.164)
*Bacteroides*	0.511 (−0.236)	0.854 (−0.067)	0.461 (−0.264)	0.802 (−0.091)	0.108 (−0.539)	0.106 (−0.541)	0.679 (−0.15)	0.615 (0.182)	0.199 (0.444)
*Bifidobacterium*	0.651 (0.164)	0.364 (0.322)	0.599 (0.19)	0.464 (−0.262)	0.31 (0.358)	0.213 (0.432)	0.365 (0.321)	0.028 (0.687)	0.855 (0.067)
*Lactobacillus*	0.851 (0.068)	0.035 (−0.668)	0.428 (−0.283)	0.228 (−0.419)	0.57 (−0.205)	0.903 (0.045)	0.379 (0.313)	0.853 (−0.068)	0.672 (0.153)
*Staphylococcus*	0.336 (0.34)	0.662 (0.159)	0.001 (0.862)	0.353 (0.329)	0.06 (0.612)	0.097 (0.553)	0.211 (0.433)	0.293 (0.37)	0.131 (0.511)
*Veillonella*	0.551 (0.215)	0.046 (−0.64)	0.878 (0.056)	0.865 (−0.062)	0.906 (0.043)	0.598 (−0.191)	0.385 (−0.309)	0.097 (−0.553)	0.385 (−0.309)

* Calprotectin (CALPR).

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
