# Peer review of "The Effectiveness of Multi-Session FMT Treatment in Active Ulcerative Colitis Patients: A Pilot Study"

_biomedicines, 2020, doi:10.3390/biomedicines8080268_

Round 1

Reviewer 1 Report

FMT could be considered the brand new frontier of IBD therapy. The effectiveness of probiotics in reducing gut inflammation and disease activity in IBD patients, is already an established fact. Introduction should include the relevant reference by Tomasello G et al: "From gut microflora imbalance to mycobacteria infection: Is there a relationship with chronic intestinal inflammatory diseases?" (2011). In the presented work, Authors administrated healthy subject microbiota to IBD patient, in order to modify their intestinal bacterial flora and lower flogosis indices. The use of human feaces allows undoubtely to prepare the right bacterial blend, but, unfortunately, it remains an invasive procedure. Several works describe the effectiveness of oral probiotics in reducing inflammation and disease activity. Please explain why FMT could be better than oral probiotcs administration and comment the article "The long-term effects of probiotics in the therapy of ulcerative colitis: A clinical study" by Palumbo VD et al (2016). A comparation with current commercial oral preparations could be interesting. Notwithstanding this, the work is well designed and very well described. It can be accepted after suggested revisions.

Author Response

Dear Reviewer,

We thank you very much for the comments and suggestions. The comments and suggestions are valuable and very helpful for revising and improving our manuscript. We have made point-by-point revisions according to the referees’ comments and suggestions, as described in the authors’ response.

Comment 1: Introduction should include the relevant reference by Tomasello G et al: "From gut microflora imbalance to mycobacteria infection: Is there a relationship with chronic intestinal inflammatory diseases?" (2011).

Answer 1 : We would like to thank you for recommending us the relevant reference by Tomasello G. et al: “From gut microflora imbalance to mycobacteria infection: Is there a relationship with chronic intestinal inflammatory diseases?” (2011). According to your comments, we included this publication in our introduction ( lines 54-64) , by explaining the most important mechanisms of dysbiosis in patients with IBD.

Comment 2: Please explain why FMT could be better than oral probiotcs administration and comment the article "The long-term effects of probiotics in the therapy of ulcerative colitis: A clinical study" by Palumbo VD et al (2016). A comparation with current commercial oral preparations could be interesting.

Answer 2: Gut microbiota dysbiosis is a condition related with the pathogenesis of vary intestinal illnesses and extra-intestinal illnesses. Dysbiosis status has been related to various important pathologies, and many therapeutic strategies aimed at restoring the balance of the intestinal ecosystem have been implemented. These strategies include the administration of probiotics, prebiotics, synbiotics; phage therapy and finally fecal transplantation. Probiotics can be used both to prevent the onset of dysbiosis when the patient is exposed to predisposing conditions and as therapeutic agents to rebalance an ongoing condition of dysbiosis. Beneficial effects of probiotic strains can be categorized as immunological and nonimmunological. A probiotic usually contains one or more species of microorganisms that need to find fertile ground to multiply. They do not permanently colonize the intestines, and adhesion to the intestinal epithelium is only temporary. After probiotic supplementation is completed, the probiotic strains are no longer detected in the stool. The use of probiotics as an adjunct to treatment is sometimes not a sufficient way to restore the intestinal homeostasis. An example is the use of probiotics in C. difficile infections that do not

produce satisfactory results.

In FMT, microbes are transplanted with their environment, and therefore, perfectly adapted to the conditions prevailing in the intestines. After FMT, the time of

restoring intestinal homeostasis is much shorter than that in which the spontaneous restoration of the microbiome devastated during the treatment process or after the use of probiotics takes place. Thorough studies on FMT, such as accurate trials and cohort studies with control groups, are desirable to corroborate its long-term effectiveness and safety.

 According to the second recommended study: ” The long-term effects of probiotics in the therapy of ulcerative colitis: A clinical study”; by Palumbo VD et al. (2016) authors present the evaluation of the long-term (2 years) effects of combination therapy (mesalazine plus a probiotic blend of Lactobacillus salivarius, Lactobacillus acidophilus, and Bifidobacterium bifidus strain BGN4) on ulcerative colitis activity. In the study presented by Palumbo VD et al., during the 24 months of study patients treated with mesalazine and probiotic blend showed better results than those reached by patients treated with mesalazine alone. From my clinical experience I have been recommending patients with chronic ulcerative colitis to use certain probiotics to support the effect of mesalazine. The efficacy of this combination is presented by obtaining a better clinical response, significantly reduced stool frequency and quality, decreasing abdominal distention, and improving the quality of life.

Comment 3: Notwithstanding this, the work is well designed and very well described. It can be accepted after suggested revisions.

Answer 3: We would like thank you for the positive feedback.

Once again, we thank you for the time you put in reviewing our paper and look forward to meeting your expectations. 

Sincerely,

Ass. Prof. Dorota Mańkowska-Wierzbicka, MD, PhD

Department of Gastroenterology, Metabolic Diseases, Internal Medicine and Dietetics

Heliodor Swiecicki Clinical Hospital

Poznan University of Medical Sciences

Przybyszewskiego 49, 60-355 Poznan, Poland

Telephone: +48601793457

Fax: +48618691686

Reviewer 2 Report

This an open pilot study in which patient selection is crucial to evaluate the clinical effect. Of 10 patients 60% improved as judged by the disease activity index and the biochemical parameters. It is, however, not stated whether the patients were chronically active or had a relapse: mean duration since onset was 5.9 years. All 10 patients took aminosalicylate and 4 took immunomodulatory treatment and the dose had not been changed the last 4 weeks. 4 weeks is a rather short time if 5-ASA dose had been increased or immunomodulatory treatment started due to a relapse. Regarding corticosteroid treatment the authors only state that the the dose was decreased by 4-5 mg/d/week (I assume of prednisolone), but it is not clearly staeted when the 4 patients that were on corticosteroids started this treatment. The high value for WBC before the fecal transplantation may indicate a corticosteroid effect; it is higher than the disease usually causes. 

The changes observed in the microbiome are of interest, since information obtained by using different protocols, in this case one colonoscopic and five duodenoscopic transfers, are of value. 

The discussion is wordy and difficult to read. It can be abbreviated to about two thirds of its length and betetr focused.

There are several linguistic errors that need to be corrected although the paper is generally not poorly written.

Author Response

Dear Reviewer,

We thank you very much for the comments and suggestions. The comments and suggestions are valuable and very helpful for revising and improving our manuscript. We have made point-by-point revisions according to the referees’ comments and suggestions, as described in the authors’ response.

Comment 1: This an open pilot study in which patient selection is crucial to evaluate the clinical effect. Of 10 patients 60% improved as judged by the disease activity index and the biochemical parameters. It is, however, not stated whether the patients were chronically active or had a relapse: mean duration since onset was 5.9 years.

Answer 1 : The mean duration of Ulcerative Colitis disease (from the first diagnosis to the FMT

treatment) was 5.9 years. During this time, patients had periods of remission and exacerbation. Patients enrolled in our study had clinically,  and endoscopically active ulcerative colitis (nine of them had severe and one moderate exacerbation).

Comment 2: All 10 patients took aminosalicylate and 4 took immunomodulatory treatment and the dose had not been changed the last 4 weeks. 4 weeks is a rather short time if 5-ASA dose had been increased or immunomodulatory treatment started due to a relapse.

Answer 2: All of our patients were chronically treated with 5-ASA, and due to the disease exacerbation, they were receiving a maximal dose of 4g daily. This dose had not been changed in the last 4 weeks. Four of our patients were on immunosuppressive therapy. Three of them were receiving azathioprine in the dosage: 2-2,5mg/kg bw/d, and one person was receiving 6-mercaptopurine in the dosage of 1-1,5mg/kg bw/d. The immunosuppressive treatment (thiopurine) was conducted for ≥3 months preceding and during the enrolment, and the dose had been stable for 4 weeks.

Comment 3: Regarding corticosteroid treatment the authors only state that the the dose was decreased by 4-5 mg/d/week (I assume of prednisolone), but it is not clearly staeted when the 4 patients that were on corticosteroids started this treatment.

Answer 3: Four of our patients were receiving oral methylprednisolone therapy for 6 weeks before the FMT administration, including the last two weeks during which the dosage had been ≤20 mg daily, and had been stable for two weeks.  For oral methylprednisolone we did a mandatory taper of up to 4mg/week so that patients would be steroid-free by week 7, which coincided with the last administration of FMT. During the study, patients remained on the same dose of 5-ASA  and thiopurine.

Comment 4:  The high value for WBC before the fecal transplantation may indicate a corticosteroid effect; it is higher than the disease usually causes. 

Answer 4: Glucocorticosteroids are known to increase the WBC count. Usually leukocytosis caused by steroid therapy reaches a maximal values within the first two weeks, after which the WBC count is decreasing. WBC count of patients enrolled in our study, who were treated with methylprednisolone was below 10,5 x 10 9 /l. However, in our study group there were patients who did not use steroids therapy and had significant leukocytosis which decreased after FMT treatment.

Comment 5: The changes observed in the microbiome are of interest, since information obtained by using different protocols, in this case one colonoscopic and five duodenoscopic transfers, are of value. 

Answer 5: We would like to thank you for the positive feedback.

Comment 6: The discussion is wordy and difficult to read. It can be abbreviated to about two thirds of its length and betetr focused.

Answer 6: We have made substantial changes in several part of the paper to address the editors’ comments. Discussion ( lines 143-286) has been abbreviated and better focused.

Comment 7: There are several linguistic errors that need to be corrected although the paper is generally not poorly written.

Answer 7: We have corrected the linguistic errors in our manuscript.

Once again, we thank you for the time you put in reviewing our paper and look forward to meeting your expectations. 

Sincerely,

Ass. Prof. Dorota Mańkowska-Wierzbicka, MD, PhD

Department of Gastroenterology, Metabolic Diseases, Internal Medicine and Dietetics

Heliodor Swiecicki Clinical Hospital

Poznan University of Medical Sciences

Przybyszewskiego 49, 60-355 Poznan, Poland

Telephone: +48601793457

Fax: +48618691686

Round 2

Reviewer 2 Report

The authors have considered the criticisms from the reviewers. To evaluate the treatment effect of the FTM in this kind of open pilot study of UC is hazardous, although the authors have now provided better information on the concommittant treatment. It may be a too strong statement to say "...resulted in remission of UC" in the last sentence in the abstract.